# A Fatal Case of Neuroblastoma Complicated by Posterior Reversible Encephalopathy with Rapidly Evolving Transplantation-Associated Thrombotic Microangiopathy

**DOI:** 10.3390/children10030506

**Published:** 2023-03-03

**Authors:** Motohiro Matsui, Atsushi Makimoto, Yuya Saito, Mikako Enokizono, Kentaro Matsuoka, Yuki Yuza

**Affiliations:** 1Department of Hematology/Oncology, Tokyo Metropolitan Children’s Medical Center, Tokyo 183-8561, Japan; 2Division of Molecular Epidemiology, Jikei University School of Medicine, Tokyo 105-8461, Japan; 3Department of Laboratory Medicine, Tokyo Metropolitan Children’s Medical Center, Tokyo 183-8561, Japan; 4Department of Pediatrics, Tama-Hokubu Medical Center, Tokyo 189-8511, Japan; 5Department of Radiology, Tokyo Metropolitan Children’s Medical Center, Tokyo 183-8561, Japan; 6Department of Pathology, Tokyo Metropolitan Children’s Medical Center, Tokyo 183-8561, Japan

**Keywords:** transplantation-associated thrombotic microangiopathy (TA-TMA), bone marrow transplantation (BMT), progressive posterior reversible encephalopathy (PRES), neuroblastoma, children

## Abstract

Background: Transplantation-associated thrombotic microangiopathy (TA-TMA) is a severe complication of hematopoietic stem cell transplantation and is sometimes fatal. Observations: A 4-year-old, male patient with stage M neuroblastoma (NBL) who had received an allogeneic bone marrow transplantation (BMT) from his sibling five months previously presented with rapidly progressive posterior reversible encephalopathy (PRES) complicated with TA-TMA. Although the patient was transferred to the pediatric intensive care unit, he died within one week after the onset of the latest symptoms. Conclusions: This is the first description of a fatal case of NBL complicated by PRES with rapidly evolving TA-TMA after an allogenic BMT.

## 1. Introduction

Transplantation-associated thrombotic microangiopathy (TA-TMA) is a severe complication of hematopoietic stem cell transplantation and is sometimes fatal. TA-TMA is defined as a syndrome of microangiopathic hemolytic anemia, renal and neurological dysfunction [1]. The pathophysiology of TA-TMA is initiated by endothelial injury and microthrombi formation [2], most commonly affecting the kidneys, lungs, and the gastrointestinal tract [3]. Among patients with TA-TMA, the most common cause of death is organ failure (24%) whereas CNS failure (0.15%) is quite rare [4]. There are few data on CNS failure in TA-TMA, and the details remain elusive. A well-known form of TA-TMA-related CNS failure is acute, uncontrolled, TMA-associated hypertension, including posterior reversible encephalopathy (PRES), which may be accompanied by CNS bleeding [2]. 

The present report described the clinical course of a 4-year-old, male patient with neuroblastoma with TA-TMA development, which induced PRES syndrome leading rapidly to a fatal outcome. In addition, we conducted a review of past studies of this topic using medical databases to investigate the clinicopathological features of fatal CNS injury associated with TA-TMA.

## 2. Case Presentation

A 4-year-old, Japanese male patient received the diagnosis of stage M NBL based on the International Neuroblastoma Risk Group Staging System (INRG-SS) [5] originating in the posterior mediastinum and metastasizing to the bone marrow at multiple sites. A pathological analysis demonstrated unfavorable features, including a poorly differentiated subtype and a low mitotic index (MKI) according to the International Neuroblastoma Pathology Classification. MYCN was not amplified by fluorescence in situ hybridization. The patient received four courses of an induction regimen consisting of cyclophosphamide, vincristine, pirarubicin, and cisplatin [6] followed by four courses of ifosfamide plus etoposide (IE). As residual left parietal bone and right frontal bone lesions were detected by metaiodobenzylguanidine (MIBG) scintigraphy after six courses of the induction chemotherapy, radiation therapy (21.4Gy/12-fraction) for the residual disease was given concomitantly with IE therapy. No surgical resection of the primary tumor in the posterior mediastinum was performed because the patient had achieved a complete response.

At this juncture, the patient had good organ function and normal results on a neurological examination. He received an autologous peripheral blood stem cell (PBSC) transplantation with a conditioning regimen consisting of melphalan monotherapy (140 mg/m^2^) because the quantity of PBSC collected was insufficient (0.4 × 10^6^/kg CD 34-positive cells). Then, a second, allogeneic bone marrow (BM) transplantation was planned using bone marrow from his 8/8-matched sibling. A conditioning regimen consisting of fludarabine 40 mg/m^2^ on days −6 to −3 and busulfan 1.2 mg/kg on days −6 to −3 was followed by the transplantation of 2.0 × 10^8^/kg BM nucleated cells. GVHD prophylaxis consisted of tacrolimus therapy starting on day − 1 as well as methotrexate therapy on days + 1, 3, 6, and 11.

Neutrophil engraftment became evident on day 14 (ANC > 500/μL for three, consecutive days), and platelets exceeded 20,000/μL without a transfusion for one week on day 53. A donor-recipient chimeric analysis was performed with the patient’s bone marrow on days 30, 60, and 100 using variable number of tandem repeat typing. Since no acute or chronic GvHD developed, tacrolimus was tapered from day 41 post-BMT and discontinued on day 54. 

The patient had no infectious episodes between the start of myeloablative therapy and engraftment. He began isotretinoin therapy on day 68 followed by radiation therapy for the primary tumor in the posterior mediastinum (36 GyE/20 fr) and the residual metastatic lesion in the right frontal bone (14.8 Gy/8 fr). Five months after the BMT, he experienced a fever, vesicular rash, and desaturation. Varicella-zoster virus (VZV) DNA was detected in a serum sample by PCR. VZV pneumonia was diagnosed and treated with intravenous acyclovir for 14 days. Two weeks later, he presented at the outpatient clinic with a stomachache. Although his physical signs were unremarkable, he was hospitalized for possible emergencies.

On hospital day 2, the patient’s stomachache improved, but he showed facial edema and erythema of the upper extremities. On hospital day 4, he had a fever and was started cefepime. A real-time polymerase chain reaction (PCR) assay for respiratory virus which was performed with a nasal swab, showed negative results. On hospital day 6, acute right hemiplegia and aphasia developed. Emergency non-contrast head computed tomography (CT) revealed faint, low-density areas in the bilateral centrum semiovale. At night on the same day, he suddenly became unresponsive to stimulation and had ataxic respiration. Tests of cerebrospinal fluid (CSF) showed a normal range of cell counts, negative bacterial culture, and negative results of real-time PCR for virus including VZV. 

The patient was transferred to the pediatric intensive care unit (PICU) and required cardiorespiratory support with mechanical ventilation. On hospital day 7, his systolic blood pressure was mildly elevated and laboratory tests demonstrated elevated LDH (509 U/L), proteinuria, and thrombocytopenia. Serum creatinine was in the normal range (0.25 mg/dL) and there was no evidence of schistocytes on a peripheral blood smear. Thus, the patient did not meet any of the four diagnostic criteria for TA-TMA on hospital day 7 (Table 1) [3,7,8,9].

Intravenous acyclovir, methylprednisolone pulse, and intravenous immunoglobulin (IVIG) were started to cover various etiologies including VZV encephalopathy, GVHD of the central nervous system, and autoimmune enteropathy. However, on hospital day 8, he met all four of the diagnostic criteria for TA-TMA [3,7,8,9]. His systolic blood pressure continued to elevate and laboratory tests demonstrated elevated LDH (1383 U/L), proteinuria, thrombocytopenia, anemia, and peripheral schistocytes. A Coombs test returned negative, and a coagulation panel was normalized. An ADAMTS13 activity assay indicated 99% and the CH50 level was 40 U/mL (Table 2). 

Magnetic resonance imaging (MRI) of the brain on the same day demonstrated extensive vasogenic edema with multiple microinfarctions in the bilateral cerebral white matter, genu of the corpus callosum, external capsule, posterior limb of the internal capsule, midbrain, middle cerebellar peduncle, and cerebellar white matter (Figure 1). Magnetic resonance angiography (MRA) of the brain demonstrated diffuse vasoconstriction of the cerebral arteries, suggesting vascular endothelial injury. Arterial spin labeling (ASL)-cerebral blood flow (CBF) (post labeling delay: 1990 ms) demonstrated diffuse brain hypoperfusion. Proton magnetic resonance spectroscopy (1H-MRS) of the right central semiovale (TE/TR = 30/1500 ms) showed a prominent doublet peak for lactate at 1.3 ppm, a marker of anaerobic metabolism. On the same day, electroencephaloscopy demonstrated a flat electroencephalogram (EEG). The patient’s kidney function steadily worsened, peaking on hospital day 10 (BUN 67.7 mg/dL and creatinine 1.26 mg/dL). On hospital day 11, non-contrast head CT demonstrated extensive swelling of the entire brain causing downward herniation. There was no room for neurosurgical intervention. 

He finally died on hospital day 12. A CSF analysis on day 6 revealed mildly increased cytokines (INF-γ 18.53 pg/mL, IL-1β 72.89 pg/mL, IL-6 26.64 pg/mL, IL-10 82.65 pg/mL, TNF-α 28.29 pg/mL) and chemokines (CXCL13 7.13 pg/mL, CXCL1 253.38 pg/mL, CXCL10 409.29 pg/mL, CCL2 1545.2 pg/mL, MIF 4972.24 pg/mL). An autopsy revealed diffuse acute thrombotic microangiopathy in the glomeruli and arterioles primarily affecting the kidneys and significant gross brain edema. Glomerular fibrin thrombi with entrapped red blood cells were observed in the left kidney, and fibrin thrombi were also noted in the arterioles (Figure 2). CD3-positive T cell and macrophage infiltration was observed in the edematous brain tissue of the left occipital lobe.

## 3. Discussion

We reported a 4-year-old, male patient with rapidly progressive PRES complicated with TA-TMA and stage M NBL who had received an allogeneic BMT from his sibling five months previously. The patient’s condition rapidly deteriorated rapidly after the onset of his neurological symptoms, resulting in death from PRES shortly after the diagnosis of TA-TMA was made.

Although neurological defects have been reported in up to half of all patients with TA-TMA, fatal CNS injury with TA-TMA has been reported in only 0.15% of the patients [4]. A search on PubMed using the terms “transplantation-associated or hematopoietic stem cell transplantation-associated thrombotic microangiopathy and posterior reversible encephalopathy or central nervous system failure” demonstrated that, as of 31 December 2022, there was only one reported case of fatal CNS injury in an 11-year-old, female patient with Danon disease who experienced a hyper-acute complication of diffuse TA-TMA after an orthotopic heart transplantation. The patient’s course was similar to that of our patient. Shortly after the transplantation, severe kidney injury and progressive altered mental status developed, culminating in cerebral edema, brain herniation, and death [11]. There was no previous report describing TA-TMA associated with bone marrow transplantation involving fatal cerebral edema resulting from PRES.

Although PRES was initially described as a benign, reversible condition with a good prognosis, the mortality rate associated with this condition is 19% [12]. The prognosis of PRES is highly dependent on its etiology. A recent systemic review and meta-analysis including 448 patients with PRES demonstrated a good outcome in patients with PRES related to pre-eclampsia/eclampsia [13]. 

There are other factors associated with a poor prognosis. The classic imaging patterns usually reveal vasogenic edema involving the parieto-occipital region. Atypical imaging findings, including a central-variant (brainstem) pattern, unilateral involvement, restricted diffusion, intracerebral hemorrhage, and contrast enhancement, have also been described. Various imaging findings associated with a poor outcome include extensive cerebral edema or worsening severity on imaging studies, massive hemorrhage, and restrictive diffusion [14]. In our case, the rapid progression of severe, global brain edema involving the brainstem proved fatal.

Regarding the etiology of fatal cerebral edema, Takanashi et al. reported four patients with hemolytic uremic syndrome (HUS) induced by a STEC O111 infection. All the patients eventually died from progressive cerebral edema, which rapidly deteriorated within 48 hours and resulted in herniation on days 1 to 3 from onset [15]. Postmortem neuropathological examination of three of the patients revealed severe, noninflammatory cerebral edema and herniation with pathological findings similar to those of our case. The authors considered Shiga toxin and cytokine storm-induced injury to the endothelial cells negatively affecting the blood-brain-barrier (BBB) as possible causes of progressive cerebral edema with HUS [16]. In the present case, the severe varicella infection with the cytokine storm may have similarly triggered BBB disruption, which was preceded by severe endothelial cell damage caused by pretransplant treatments, including multiagent chemotherapy and irradiation. This constellation of factors may have eventually caused TA-TMA complicated with PRES and the fatal cerebral edema. 

To investigate the etiology of our case, we measured the level of ten cytokines and chemokines in CSF during the acute phase of PRES with TA-TMA. All ten cytokines and chemokines were mildly elevated compared with the control values in another study [14]. Shimizu et al. reported that the levels of proinflammatory cytokines (IL-6 and IL-8) were much higher in patients with severe HUS than in patients with mild HUS, suggesting that proinflammatory cytokines may play an important role in the pathogenesis of the condition [17]. In our case, the CSF level of IL-6 was not much higher than that seen in severe influenza encephalopathy (mean CSF IL-6 level: 453.7 ± 372.4 pg/mL) in another study [18]. This fact corroborates our hypothesis that the endothelial damage in the present case was not caused by a cytokine storm secondary to a viral infection (i.e., influenza encephalopathy) but by the pretransplantation treatment, irradiation, and severe varicella infection.

Several diagnostic guidelines for TA-TMA have been published [3,7,8,9], but they do not agree on all points. Furthermore, a diagnosis of TA-TMA is challenging, given that the patients undergoing HSCT have abnormal hematological and renal parameters for any of a number of disparate reasons. In our case, there was no time to treat the TA-TMA because the patient met the diagnostic criteria almost at the same time as when pupillary dilation and EEG-flattening were detected. Sonata et al. reported that proteinuria, systolic hypertension, and elevated LDH occurred ten to 14 days prior to TMA diagnosis, suggesting that these findings were early diagnostic markers [8]. Our case met the diagnostic criteria only one day prior to EEG-flattening. Identifying the early diagnostic markers is necessary to detect the onset of rapid fatal PRES with TA-TMA.

## 4. Conclusions

The present report is the first to describe a fatal case of neuroblastoma complicated by PRES with rapidly evolving TA-TMA after an allogenic BMT. Diagnostic criteria enabling early detection of the initial signs and symptoms are needed to identify the hyper-acute onset of fatal PRES with TA-TMA.

## Figures and Tables

**Figure 1 children-10-00506-f001:**
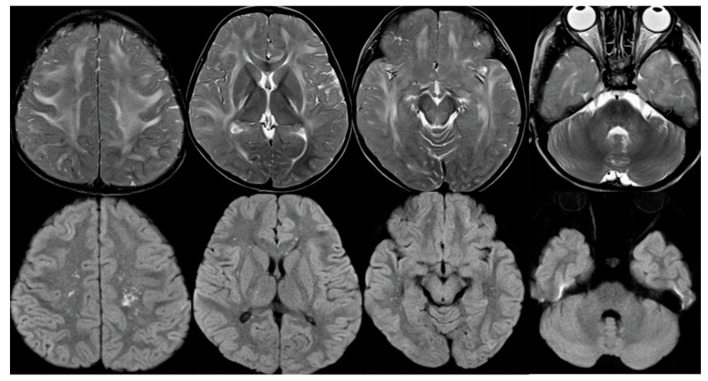
Brain MRI on hospital day 8. T2-weighted images (**upper** row) demonstrated extensive vasogenic edema in the bilateral cerebral white matter, genu of the corpus callosum, external capsule, posterior limb of the internal capsule, midbrain, middle cerebellar peduncle, and cerebellar white matter. Diffusion-weighted images (**lower** row) showed multiple microinfarctions in the bilateral cerebral white matter.

**Figure 2 children-10-00506-f002:**
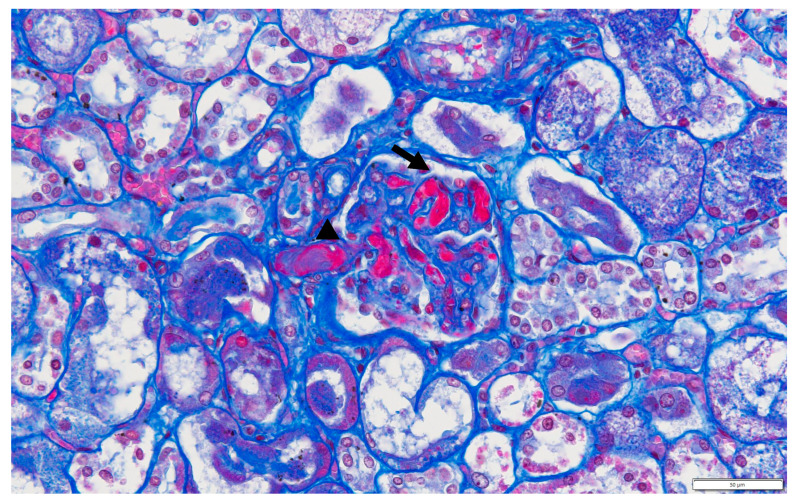
Glomerular fibrin thrombi with entrapped red blood cells can be seen (arrow). Fibrin thrombi can also be seen in the arteriole (arrowhead). (Masson trichrome ×40).

**Table 1 children-10-00506-t001:** Four diagnostic criteria for TA-TMA [10].

Parameter	CTN 2005 [9]	IWG 2007 [8]	Cho et al. [7]	Jodele et al. [3]
Schistocytes	≥2/HPF	>4%	≥2/HPF	Present
Serum LDH	Elevated	Sudden or persistent elevation	Elevated	Elevated
Renal and/or neurological dysfunction	Serum creatinine 2 × baseline or 50% dec’d creatinine clearance	NA	NA	Proteinuria ≥30 mg/dL or hypertension
Direct and indirect Coombs test	Negative	NA	Negative	NA
Thrombocytopenia	NA	De novo prolonged or progressive	De novo prolonged or progressive	De novo
Anemia	NA	Decreased Hb or increased transfusion requirements	Decreased Hb	Decreased Hb or increased transfusion requirements
Serum haptoglobin	NA	Decreased	Decreased	NA
Terminal complement activation	NA	NA	NA	Present

CTN, Blood and Marrow Transplant Clinical Trials Network; Hb, hemoglobin; HPF, high-power field; IWG, European LeukemiaNet International Working Group; LDH, lactate dehydrogenase; NA, not applicable; TA-TMA, transplantation-associated thrombotic microangiopathy.

**Table 2 children-10-00506-t002:** Laboratory findings and blood pressure of the patient.

Test (Reference Range)	Day 1	Day 5	Day 7	Day 8
Leukocytes (4200–18,500/mm^3^)	8540	8190	12,900	11,220
Hemoglobin (11.2–14.2 g/dL)	14.5	9.7	11.7	8.5*1
Platelets (180,000–550,000/mm^3^)	145,000	74,000	65,000	80,000*2
CRP (0.00–0.01 mg/dL)	7.10	3.47	2.98	3.77
AST (24–41 IU/L)	33	24	32	45
ALT (9–28 IU/L)	32	15	15	16
Creatinine (0.22–0.43 mg/dL)	0.32	0.38	0.25	0.38
LDH (185–350 IU/L)	288	304	509	1383
Haptoglobin	NA	NA	NA	<10
Peripheral schistocytes	-	-	-	+
Direct Coombs test	NA	NA	NA	Negative
CH50 (25–48/mL)	NA	NA	NA	40

CRP, C-reactive protein; AST, Aspartate aminotransferase; ALT, Alanine aminotransferase; LDH, lactate dehydrogenase; NA, not applicable. *1. After one unit red blood cell transfusion. *2. After ten units platelet cell transfusion.

## Data Availability

Not applicable.

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
