# Peer review of "A Fatal Case of Neuroblastoma Complicated by Posterior Reversible Encephalopathy with Rapidly Evolving Transplantation-Associated Thrombotic Microangiopathy"

_children, 2023, doi:10.3390/children10030506_

Round 1

Reviewer 1 Report

In this case report by Matsui et al, the authors present a case of rapidly progressive neurological failure in the setting of transplant associated TMA, a overall rare occurrence. The case report is overall well written. There are couple of suggestions to improve upon the description of the case:

Decreased haptoglobin an elevated serum creatinine were “denied”- does this mean they were not present (or normal)? Please clarify

On hospital day 8, he met all four diagnostic criteria for TMA—please describe with specific lab values how the four diagnostic criteria were met. Were CH50 levels repeated at that time and if so, what was the value?

Was the patient hypertensive during the evolution of the TMA and or PRES? No mention of their blood pressure, whether it was normal or abnormal, is given in the case presentation….

Were any fibrin thrombi present in the cerebral vasculature at the time of autopsy?

Author Response

We wish to express our appreciation to the reviewer for their insightful comments, which have helped us significantly improve the paper.

Comment 1: Decreased haptoglobin an elevated serum creatinine were “denied”- does this mean they were not present (or normal)? Please clarify

Response: Thank you for your comment. According to your suggestion, we revised the sentence ‘Serum creatinine was in normal range (0.25 mg/dL) and there was no evidence of schistocytes on a peripheral blood smear. Thus, the patient did not meet any of the four diagnostic criteria for TA-TMA on hospital day 7 (table 1) [5,6,7,8]. Intravenous acy-clovir, methylprednisolone pulse, and intravenous immunoglobulin (IVIG) were started to cover various etiologies including VZV encephalopathy, GVHD of central nervous system, and autoimmune enteropathy. However, on hospital day 8, he met all four of the diagnostic criteria for TA-TMA [5,6,7,8]. His systolic blood pressure was continued elevating and laboratory tests demonstrated elevated LDH (1383 U/L), pro-teinuria, thrombocytopenia, anemia, and peripheral schistocytes. A Coombs test re-turned negative, and a coagulation panel was normalized. ADAMTS13 activity assay indicated 99% and the CH50 level was 40 U/mL (table 2).’ in line 98-108.

Comment 2: On hospital day 8, he met all four diagnostic criteria for TMA—please describe with specific lab values how the four diagnostic criteria were met. Were CH50 levels repeated at that time and if so, what was the value?

Response: Thank you for your comment. We made a table 1 demonstrating his laboratory test results. We have not repeatedly measured CH50.

Comment 3:Was the patient hypertensive during the evolution of the TMA and or PRES? No mention of their blood pressure, whether it was normal or abnormal, is given in the case presentation….

Response: According to your suggestion, we added the sentence ‘his systolic blood pressure was mildly elevated and laboratory tests demonstrated ele-vated LDH (509 U/L), proteinuria, and thrombocytopenia. Serum creatinine was in normal range (0.25 mg/dL) and there was no evidence of schistocytes on a peripheral blood smear. Thus, the patient did not meet any of the four diagnostic criteria for TA-TMA on hospital day 7 (table 1) [5,6,7,8]. Intravenous acyclovir, methylpredniso-lone pulse, and intravenous immunoglobulin (IVIG) were started to cover various eti-ologies including VZV encephalopathy, GVHD of central nervous system, and auto-immune enteropathy. However, on hospital day 8, he met all four of the diagnostic criteria for TA-TMA [5,6,7,8]. His systolic blood pressure was continued elevating and laboratory tests demonstrated elevated LDH (1383 U/L), proteinuria,,………’ in line 96 and 106. And we made a table 1 demonstrating his systolic blood pressure.

Comment 4: Were any fibrin thrombi present in the cerebral vasculature at the time of autopsy?

Response: Fibrin thrombi was not present in the cerebral vasculature at the time of autopsy.

Reviewer 2 Report

This case is a very interesting case aiming to shine a light on a topic with, indeed, a vague presentation most times. The study is well-written and provides a comprehensive description of the patient's course; however, some additional information might be necessary. 

Firstly, there in not any information regarding the patient's course (tests performed, results, treatment) during the first five days since the patient's admission. Secondly, it would be interesting to provide a table demonstrating the patient's cbc and biochemistry lab results during his hospitalization (e.g. WBCs, Hg, hct, platelets, U, Cr, LDH, SGOT). Thirdly, it would be wise to report the diagnostic criteria of TA-TMA in the manuscript, since is the main topic of the study. 

Author Response

We wish to express our appreciation to the reviewer for their insightful comments, which have helped us significantly improve the paper.

Comment 1:Firstly, there in not any information regarding the patient's course (tests performed, results, treatment) during the first five days since the patient's admission.

Response: According to your suggestion, we added the sentence ‘On hospital day 2, his stomachache improved, but he showed facial edema and ery-thema of upper extremities. On hospital day 4, he had a fever and started cefepime. A real-time polymerase chain reaction (PCR) assay for respiratory virus which was per-formed with his nasal swab, showed negative results. On hospital day 6, acute right hemiplegia and aphasia developed. Emergency non-contrast head computed tomog-raphy (CT) revealed faint, low-density areas in the bilateral centrum semiovale. At night on the same day, he suddenly became unresponsive to stimulation and had atax-ic respiration. Tests of cerebrospinal fluid (CSF) showed normal range of cell counts, negative bacterial culture, and negative results of real time PCR for virus including VZV.’ in line 86 and 94.

Comment 2: Secondly, it would be interesting to provide a table demonstrating the patient's cbc and biochemistry lab results during his hospitalization (e.g. WBCs, Hg, hct, platelets, U, Cr, LDH, SGOT).

Response: Thank you for your comment. We made a table 2 demonstrating his laboratory test results.

Comment 3: Thirdly, it would be wise to report the diagnostic criteria of TA-TMA in the manuscript, since is the main topic of the study.

According to your suggestion, we added the table 1 showing the diagnostic criteria of TA-TMA.

Reviewer 3 Report

Dr Matsui presents an interesting case report of 4 y.o. patients' unfortunate death after complication by posterior reversible encephalopathy with rapidly evolving transplantation-associated thrombotic microangiopathy.

This is pretty interesting observation, because searching the PubMed database showed only one similar case in the past.

Since the presented draft is submitted as a case report, I found high merit and novelty in a presented work. I appreciate very detailed the course of post-transplantional phase of care of the patient. This workflow might be very useful for many clinicians. I also found all the implemented procedures reasonable and matched to state-of-the-art.

I have some comments that should be addressed during the peer review process:

- Please briefly lenghten the introduction section. Despite that this is a case report, providing more background would be beneficial for clinicians outside of the discussed field.

- Please extend the PubMed research until December 31st, 2022. July 31st 2022 is outdated. line 123

- Please provide the ethnicity of the patient.

- Please provide the list of measured cytokines.

- Please provide the Readers with short comment regarding the chances for improvement in the future cases based on what we learnt from this particular case.

- Figures should be labeled with the scales

- Figure 2 lacks the arrow

Author Response

We wish to express our appreciation to the reviewer for their insightful comments, which have helped us significantly improve the paper.

Comment 1: Please briefly lengthen the introduction section. Despite that this is a case report, providing more background would be beneficial for clinicians outside of the discussed field.

Response: According to your suggestion, we added the sentence in introduction ‘TA-TMA is defined as a syndrome of microangiopathic hemolytic anemia, renal and neurological dysfunction [18]. Pathophysiology, TA-TMA was resulted from endothelial injury and microthrombi formation [2], most commonly affecting the kidneys, lungs, and the gastrointestinal [8].’ in line 32 and 36.

Comment 2:  Please extend the PubMed research until December 31st, 2022. July 31st 2022 is outdated. line 123

Response: Thank you for your comment. we revised the manuscript.

Comment 3: Please provide the ethnicity of the patient.

Response: Thank you for your comment. we revised the manuscript in line 47.

Comment 4: Please provide the list of measured cytokines.

Response: Thank you for your comment. All measured cytokine values ​​are listed in the article.

Comment 4: Please provide the Readers with short comment regarding the chances for improvement in the future cases based on what we learnt from this particular case.

Response: Thank you for your comment. our case is a very difficult case to improve, and the only comments we could provide is ‘Identifying the early diagnostic markers is necessary to detect the onset of rapid fatal PRES with TA-TMA.’ in line 195-196.

Comment 5: Figures should be labeled with the scales

Figure 2 lacks the arrow

Response: Thank you for your comment. We revised the Figures.

Round 2

Reviewer 3 Report

The Authors correctly addressed raised majors. I have no further comments.